# UniRelight: Learning Joint Decomposition and Synthesis for Video Relighting

**Kai He**[1,2,3]     **Ruofan Liang**[1,2,3]     **Jacob Munkberg**[1]     **Jon Hasselgren**[1]

**Nandita Vijaykumar**[2,3]     **Alexander Keller**[1]     **Sanja Fidler**[1,2,3]

**Igor Gilitschenski**[2,3†]     **Zan Gojcic**[1†]     **Zian Wang**[1,2,3†]

[1]NVIDIA     [2]University of Toronto     [3]Vector Institute

## Abstract

We address the challenge of relighting a single image or video, a task that demands precise scene intrinsic understanding and high-quality light transport synthesis. Existing end-to-end relighting models are often limited by the scarcity of paired multi-illumination data, restricting their ability to generalize across diverse scenes. Conversely, two-stage pipelines that combine inverse and forward rendering can mitigate data requirements but are susceptible to error accumulation and often fail to produce realistic outputs under complex lighting conditions or with sophisticated materials. In this work, we introduce a general-purpose approach that jointly estimates albedo and synthesizes relit outputs in a single pass, harnessing the generative capabilities of video diffusion models. This joint formulation enhances implicit scene comprehension and facilitates the creation of realistic lighting effects and intricate material interactions, such as shadows, reflections, and transparency. Trained on synthetic multi-illumination data and extensive automatically labeled real-world videos, our model demonstrates strong generalization across diverse domains and surpasses previous methods in both visual fidelity and temporal consistency. Our project page is https://research.nvidia.com/labs/toronto-ai/UniRelight/.

## 1  Introduction

Lighting is crucial in defining a scene's visual appearance, influencing both aesthetics and the perception of objects and materials. Modifying illumination while preserving scene content enables a wide range of applications, including creative editing, simulation, and robust vision systems. Realistic relighting necessitates an accurate understanding of scene geometry and materials, as well as faithful light transport modeling. Recent methods aim to learn these complex relationships from data, but a significant challenge is the scarcity of multi-illumination datasets capturing the same scene under varied lighting conditions. Acquiring such data in the real world requires controlled environments, calibrated equipment, and repeated captures, making it costly and impractical. As a result, most existing approaches rely on synthetic data or focus on narrow domains like portraiture, limiting their ability to generalize to real-world scenes.

To address the lack of multi-illumination data, recent works [69, 39] have decomposed the relighting task into two stages: an inverse rendering step that estimates scene attributes such as albedo, normal, and depth (i.e., G-buffers), followed by a forward rendering step that synthesizes relit images conditioned on these estimates. This pipeline avoids the need for multi-illumination supervision, allowing each stage to be trained separately using single-illumination data. However, inverse and forward rendering are inherently coupled, and modeling them separately with discrete intermediate representations introduces significant limitations. The forward renderer is highly sensitive to errors in

---

† Joint Advising

39th Conference on Neural Information Processing Systems (NeurIPS 2025).

Input & Albedo Pred.          Relighting I                    Relighting II

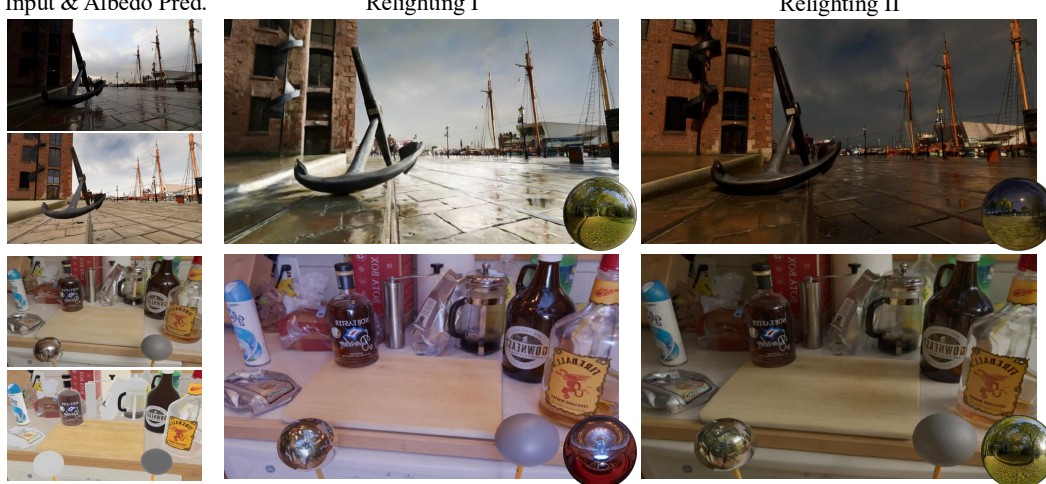

Figure 1: Given an input image (top left) or video, our method *jointly* estimates albedo (bottom left) and synthesizes relit videos with novel lighting conditions using provided HDR probes. Notably, our estimated albedo maps effectively demodulate shadows and specular highlights, while the relit images exhibit plausible shadows and specular highlights.

the estimated G-buffers and struggles to capture complex material properties not encoded by them, such as transparency and subsurface scattering.

Learning an end-to-end relighting model in low data regime is challenging. While prior methods for intrinsic estimation (e.g., geometry [28, 19] and albedo [32, 39]) trained on synthetic data often generalize well to real-world scenes, we observe that directly training relighting models as video-to-video translation leads to poor generalization to unseen domains. We argue that robust relighting requires an explicit understanding of illumination and scene properties.

Hence, we propose a relighting framework that *jointly* models the distribution of scene intrinsics and illumination. Inspired by VideoJAM [12], we train a video generative model that *jointly* denoises latent space for relighting and albedo demodulation in a single pass. Practically, we concatenate both latent representations and treat them as a single video clip. This design is motivated by the hypothesis that demodulation provides a strong prior for the relighting task, such as removing shadows. This *joint* formulation encourages the model to learn an internal representation of scene structure, leading to improved generalization across diverse and unseen domains. Our approach contrasts with two-stage pipelines that rely on explicit G-buffer estimation. Instead, we implicitly reason about intrinsic scene representation, enabling better representation learning, reducing error accumulation, and modeling of complex visual effects such as specular highlights, transparency, and subsurface scattering.

Our formulation also offers flexibility to learn from diverse sources of supervision. We leverage high-quality synthetic data for full supervision and complement it with real-world single illumination videos that can be auto-labeled at scale. This hybrid training strategy enables the model to handle complex lighting effects while significantly improving generalization to unseen domains, see Figure 1. UNIRELIGHT enables high-quality relighting and intrinsic decomposition from a single input image or video, producing temporally consistent shadows, reflections, and transparency, and outperforms state-of-the-art methods.

## 2 Related Work

**Inverse rendering** estimates intrinsic scene properties such as geometry, materials, and lighting from input images. Traditional approaches use hand-crafted priors within an optimization framework [34, 3, 21, 11, 77, 2] to handle low-order effects like diffuse shading and reflectance. Recently, the field has been revitalized by applying supervised and self-supervised learning [4, 33, 35, 57, 66, 36, 37, 10, 61, 60, 63, 6] to inverse rendering tasks. These methods typically require large, domain-specific datasets, and struggle to generalize outside the training domain.

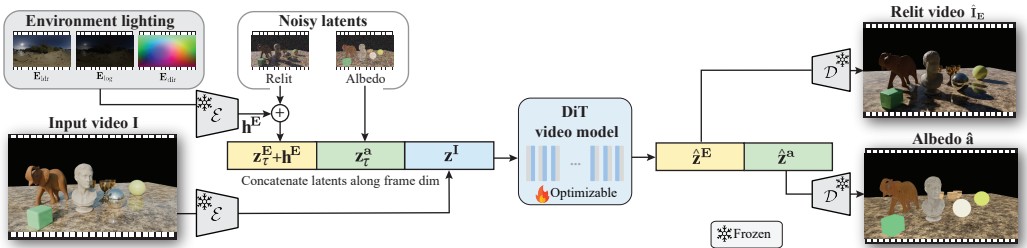

Figure 2: **Method overview.** Given an input video $\mathbf{I}$ and a target lighting configuration $(\mathbf{E}_{\text{ldr}}, \mathbf{E}_{\text{log}}, \mathbf{E}_{\text{dir}})$, our method jointly predicts a relit video $\hat{\mathbf{I}}_E$ and its corresponding albedo $\hat{\mathbf{a}}$. We use a pretrained VAE encoder-decoder pair $(\mathcal{E}, \mathcal{D})$ to map input and output videos to a latent space. The latents for the target relit video and albedo are concatenated along the temporal (frame) dimension with the encoded input video. Lighting features $\mathbf{h}^{\mathbf{E}}$, derived from the environment maps, are concatenated along the channel dimension with the relit video latent. A finetuned DiT video model denoises the joint latent according to Equation 1, enabling consistent generation of both relit appearance and intrinsic decomposition.

Most related to our approach are methods that leverage large generative image and video models for inverse rendering [18, 32, 54, 40, 30, 14, 47, 41]. Notably, RGB↔X [69] demonstrates both intrinsic decomposition and neural rendering from intrinsics using finetuned image diffusion models. The DiffusionRenderer [39] extends RGB↔X to video and also supports relighting.

**Relighting** is the task of modifying the lighting conditions in images or videos. Many methods first reconstruct 3D scenes from multi-view images [46, 22, 13, 9, 71, 76, 70, 56, 62, 38, 64, 42, 58, 25]. Previously, surface reflectance has been captured using calibrated light stages [15]. Now, relighting is supported by material properties recovered from inverse rendering. These methods typically require per-scene optimization and are limited to static, object-centric scenes with single illumination. Training across multiple scenes has led to the exploration of latent feature learning [78, 43, 72, 7], often incorporating neural rendering modules with intrinsic buffers as priors [53, 20, 50, 31, 65].

Recent approaches [55, 31, 68, 26, 39, 5, 75, 45] leverage diffusion models for relighting tasks. These methods are often domain-specific, such as portraits, single objects, and outdoor scenes. Despite promising results, a significant challenge remains the need for multi-illumination datasets [48], which are difficult to capture at scale.

**Joint generative modeling** approaches enable diffusion models to predict multiple modalities. Matrix3D [44] predicts pose estimation, depth, and novel view synthesis using a single DiT [52] model. VideoJAM [12] extends this by predicting both generated pixels and their corresponding motion from a single DiT. We leverage this approach to jointly predict a relit image and albedo.

## 3    Preliminaries: Video Diffusion Models

Diffusion models approximate a data distribution $p_{\text{data}}(\mathbf{I})$ by learning to iteratively denoise samples corrupted by Gaussian noise [59, 23, 17]. For efficiency, most video diffusion models (VDMs) operate in a lower-dimensional latent space [8, 1]. Given an RGB video $\mathbf{I} \in \mathbb{R}^{L \times H \times W \times 3}$ consisting of $L$ frames at resolution $H \times W$, a pre-trained VAE encoder $\mathcal{E}$ encodes the video into a latent tensor $\mathbf{z} = \mathcal{E}(\mathbf{I}) \in \mathbb{R}^{l \times h \times w \times C}$. Then, the final video $\hat{\mathbf{I}}$ is reconstructed by decoding $\mathbf{z}$ with a pre-trained VAE decoder $\mathcal{D}$. Both the training and inference stages of the VDM are conducted in this latent space.

In this work, we fine-tune a recent Diffusion Transformer (DiT) video model, Cosmos-Predict1 [1]. Encoding and decoding to and from latent space are performed by the pre-trained VAE `Cosmos-1.0-Tokenizer-CV8x8x8`, which compresses the video by a factor of eight along the spatial and temporal dimensions: $l = \frac{L}{8}$, $C = 16$, $h = \frac{H}{8}$, and $w = \frac{W}{8}$. The base model supports text- and image-guided video generation at a resolution of $704 \times 1280$ pixels.

To train the VDM, noisy versions $\mathbf{z}_\tau = \alpha_\tau \mathbf{z}_0 + \sigma_\tau \epsilon$ are constructed by adding Gaussian noise $\epsilon$, with the noise schedule provided by $\alpha_\tau$ and $\sigma_\tau$ following the EDM [27]. The diffusion model parameters $\theta$ of the denoising function $\mathbf{f}_\theta$ are optimized using the denoising score matching objective [27]. Once trained, iteratively applying $\mathbf{f}_\theta$ to a sample of Gaussian noise will produce a sample of $p_{\text{data}}(\mathbf{I})$.

# 4 Method

We propose a generative relighting framework by fine-tuning a DiT-based video diffusion model [52, 1], which jointly predicts the albedo and relit appearance from an input image or video under a target lighting condition.

At the core of our method is a joint denoising architecture: the latent representations for albedo and relit video are concatenated at the *token* level and denoised in a single pass using the DiT model. This formulation enables cross-modal interaction via self-attention, allowing the model to capture shared scene structure and improve generalization and temporal consistency.

We leverage a combination of synthetic datasets, multi-illumination data, and auto-labeled real-world data to train our model, ensuring robust generalization and high-quality output. In the following sections, we detail the model architecture, data strategy, and training objectives.

## 4.1 Model Design

Given an input video $\mathbf{I}$ and a temporally-varying target lighting condition $\mathbf{E} \in \mathbb{R}^{L \times H \times W \times 3}$ of the same dimensions, our goal is to train a model $\mathbf{f}_\theta$ that *jointly* denoises the albedo $\mathbf{a}$ of the input video and a relit video $\mathbf{I_E}$ under the target illumination $\mathbf{E}$.

As illustrated in Figure 2, the model comprises a VAE encoder-decoder pair, denoted $(\mathcal{E}, \mathcal{D})$, and a transformer-based denoising function, $\mathbf{f}_\theta$. We use the VAE encoder $\mathcal{E}$ to separately encode the input video $\mathbf{I}$, the albedo $\mathbf{a}$, and the relit video $\mathbf{I_E}$, producing the corresponding latent tensors $(\mathbf{z^I}, \mathbf{z^a}, \mathbf{z^E})$.

Unlike VideoJAM [12], which compresses a pair of latents $(\mathbf{z}^x, \mathbf{z}^y)$ using a linear layer, we find this approach does not yield acceptable quality for our applications. We adopt a simple yet effective strategy: concatenating the latents $\mathbf{z^I}$, $\mathbf{z^a}$, and $\mathbf{z^E}$ along the temporal (frame) dimension. This formulation enables the DiT model to apply full self-attention across input, albedo, and relit frames, facilitating cross-modal information exchange.

**Token embeddings.** To distinguish between the three modalities in the concatenated sequence, we combine standard RoPE positional embeddings with dedicated type embeddings. We first apply the same positional embedding to each of the three video clips, encoding the spatial-temporal position of tokens within each clip. To indicate modality, we introduce a learnable *type embedding* $\mathbf{c}_{\text{emb}} \in \mathbb{R}^{K_{\text{emb}} \times C_{\text{emb}}}$, where $K_{\text{emb}} = 3$ denotes the number of video types. Each type embedding is broadcast across spatial-temporal dimensions and concatenated along the channel dimension with its corresponding latent representation: $\mathbf{c}_{\text{emb}}^0$ is appended to $\mathbf{z^E}$, $\mathbf{c}_{\text{emb}}^1$ to $\mathbf{z^a}$, and $\mathbf{c}_{\text{emb}}^2$ to $\mathbf{z^I}$.

**Condition mask.** We attach a binary mask to each frame indicating whether it should be treated as a condition (e.g., input source video) or a denoising target (e.g., relit video). During training, we randomly vary which modalities are used as inputs, allowing the model to generalize under partial supervision. For instance, we may randomly treat $\mathbf{a}$ as either a conditioning input or a prediction target (see Section 4.3 for details).

**Encoding HDR lighting.** Environment maps contain high dynamic range (HDR) values that often exceed the intensity range of standard images, while the VAEs used in latent diffusion models are trained on low dynamic range (LDR) inputs and cannot directly encode these high-intensity signals.

To address this, we follow prior work [26, 39] and represent the lighting condition using three complementary buffers: (1) an LDR panorama $\mathbf{E}_{\text{ldr}}$ obtained via standard Reinhard tonemapping; (2) a normalized log-intensity map $\mathbf{E}_{\text{log}} = \log(\mathbf{E} + 1)/E_{\text{max}}$, where $E_{\text{max}}$ is the maximum log intensity; and (3) a directional encoding $\mathbf{E}_{\text{dir}} \in \mathbb{R}^{L \times H \times W \times 3}$, where each pixel stores a unit vector indicating its direction in the camera coordinate system. These representations are passed through the VAE encoder, yielding $\mathbf{h^E} = (\mathcal{E}(\mathbf{E}_{\text{ldr}}), \mathcal{E}(\mathbf{E}_{\text{log}}), \mathcal{E}(\mathbf{E}_{\text{dir}}))$ which is concatenated along the channel dimension with $\mathbf{z^E}$. We also append a binary condition mask to indicate whether lighting features are present. For input video and albedo tokens, we use zero-padded placeholders to maintain shape consistency.

## 4.2 Data Strategy

Similar to prior work [39], we combine a large-scale synthetic dataset with smaller real-world datasets. The synthetic dataset offers a large volume of high-quality data, which, when combined with

powerful diffusion models, shows impressive generalization to unseen domains [28, 19]. However, using synthetic target images biases the model output, resulting in a rendered look. Therefore, we also utilize smaller and lower-quality datasets of automatically labeled real-world data.

**Synthetic data curation.** Our synthetic dataset consists of rendered video clips that are tuples $(\mathbf{I}, \mathbf{I_E}, \mathbf{a}, \mathbf{E})$ of input video, relit result, albedo, and environment map. To create a large dataset with complex and diverse scenes and lighting conditions, we follow the methodology of the Diffusion-Renderer [39] and procedurally generate simple scenes with randomized environment map lighting. We curate a collection of 36,500 3D objects from the Objaverse LVIS subset, 4,260 PBR materials, and 766 HDRI environment maps, gathered from publicly available resources. Each scene contains a ground plane with a randomly selected material, and up to three randomly placed 3D objects. We perform collision detection to avoid intersecting objects. Additionally, we place up to three primitive objects (cube, sphere, and cylinder) with randomized shapes and materials, to increase variety. A randomly selected HDR environment map illuminates each scene. We generate videos with random motions including camera orbits, camera oscillation, lighting rotation, object rotation and translation. Each scene is rendered with two different illuminations under the same motion.

We rendered the dataset using a custom path tracer based on OptiX [51] at high sample counts with path length three to capture global illumination effects, producing a total of 108k videos with 57 frames per video at a resolution of $704 \times 1280$ pixels.

**MIT multi-illumination labeling.** The multi-illumination dataset of Murmann et al. [48] consists of 985 real-world indoor scenes for training and 30 scenes for testing, each lit under 25 different conditions. Each image contains a reflective chrome sphere that has been isolated and exported as an HDR image, which can be easily remapped to the latitude-longitude representation expected by our model. We create a dataset by randomly selecting pairs from the 25 lighting conditions and extracting $(\mathbf{I}, \mathbf{E}, \mathbf{I_E})$.

To obtain pseudo-groundtruth albedo labels $\mathbf{a}$, we re-implement the inverse renderer from Diffusion-Renderer [39] based on the `Cosmos-1.0-Diffusion-7BVideo2World` model [49], fine-tuned on our synthetic dataset. The resulting model produces high-quality albedo estimates across diverse images and videos. For each scene, we estimate albedo under all 25 lighting conditions and average the results to obtain a stable albedo map.

**Real-world auto-labeling.** While multi-illumination datasets provide effective supervision for relighting, they are costly to acquire and limited in scale. In contrast, the Internet offers an abundance of high-quality videos captured under single illumination. To leverage this resource, we curate 150k real-world video clips, each with 57 frames at a resolution of 704×1280, and automatically annotate them with albedo maps to create paired data $(\mathbf{I}, \mathbf{a})$ using the reproduced inverse rendering model. These auto-labeled RGB–albedo pairs significantly increase the diversity and realism of our training set, enabling improved generalization to real-world scenes.

### 4.3   Training

Our model is trained on a combination of the synthetic video dataset, the MIT multi-illumination dataset, and real-world auto-labeled data. For image datasets, we treat images as single-frame videos. For both the synthetic video dataset and the MIT multi-illumination dataset, each data sample consists of an input video $\mathbf{I}$, its corresponding albedo $\mathbf{a}$, a new environment map $\mathbf{E}$, and the target relit video $\mathbf{I_E}$ under this new illumination. The target latent variable for these datasets is constructed by concatenating the latent of the relit video, $\mathbf{z}_0^{\mathbf{E}}$, and the corresponding albedo, $\mathbf{z}_0^{\mathbf{a}}$, along the temporal dimension. Noise is introduced independently to both $\mathbf{z}_0^{\mathbf{E}}$ and $\mathbf{z}_0^{\mathbf{a}}$ to produce $\mathbf{z}_\tau^{\mathbf{E}}$ and $\mathbf{z}_\tau^{\mathbf{a}}$. The model parameters are optimized by minimizing the objective function:

$$\hat{\mathbf{z}}^{\mathbf{E}}(\theta), \hat{\mathbf{z}}^{\mathbf{a}}(\theta) \quad = \quad \mathbf{f}_\theta([\mathbf{z}_\tau^{\mathbf{E}} + \mathbf{h}^{\mathbf{E}}, \mathbf{z}_\tau^{\mathbf{a}}, \mathbf{z}^{\mathbf{I}}]; \mathbf{c}_{\text{emb}}, \tau) \tag{1}$$

$$\mathcal{L}(\theta) \quad = \quad \mathbb{E}_{(\mathbf{z}_0^{\mathbf{E}}, \mathbf{z}_0^{\mathbf{a}}) \sim p_{\text{data}}, \epsilon \sim \mathcal{N}(0, \sigma^2 I)} \left[ \left\| \hat{\mathbf{z}}^{\mathbf{E}}(\theta) - \mathbf{z}_0^{\mathbf{E}} \right\|_2^2 + \lambda_{\mathbf{a}} \left\| \hat{\mathbf{z}}^{\mathbf{a}}(\theta) - \mathbf{z}_0^{\mathbf{a}} \right\|_2^2 \right], \tag{2}$$

where $[\cdot]$ denotes concatenation in the temporal dimension, and $\lambda_{\mathbf{a}} = 0.1$ is a scalar weight for albedo.

**Training strategy.** Our training process incorporates specific conditioning strategies tailored to the data source. For synthetic data and the MIT multi-illumination dataset, we apply three different

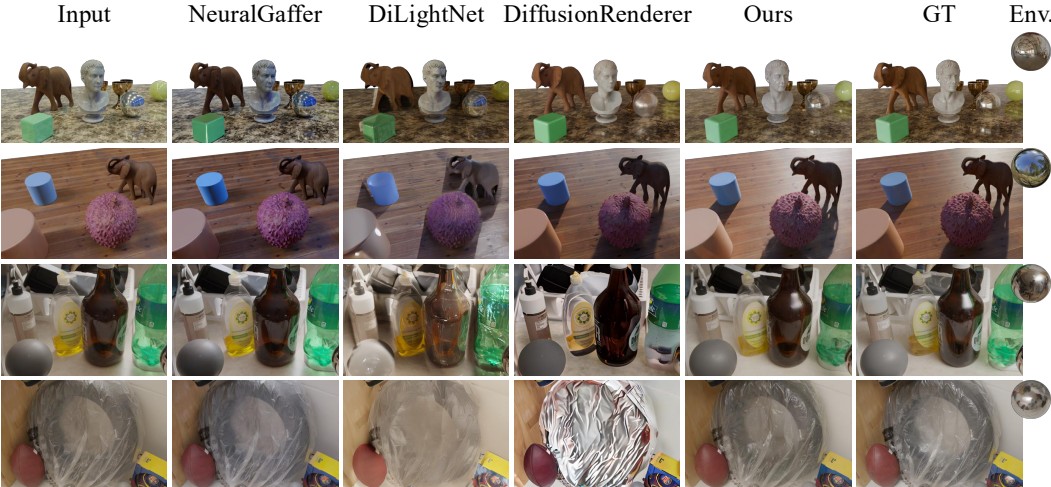

| Input | NeuralGaffer | DiLightNet | DiffusionRenderer | Ours | GT | Env. |

Figure 3: **Qualitative comparison on the synthetic dataset and MIT multi-illumination dataset.** Our method produces high-quality inter-reflections and shadows in synthetic scenes (top rows). Crucially, on the MIT multi-illumination dataset (bottom rows), it delivers relighting results with higher accuracy than baselines, which fail when faced with complex materials.

conditioning strategies. In $12\%$ of training steps, the input video is dropped (i.e., $\mathbf{z^I} = \mathbf{0}$), and the ground truth albedo serves as the sole conditioning signal. The denoising function is then $\mathbf{f}_\theta([\mathbf{z}_\tau^\mathbf{E} + \mathbf{h^E}, \mathbf{z}_0^\mathbf{a}, \mathbf{0}]; \mathbf{c}_{\text{emb}}, \tau)$. In another $18\%$ of the training steps, the model is conditioned on both the input video $\mathbf{z^I}$ and the ground truth albedo, resulting in the function $\mathbf{f}_\theta([\mathbf{z}_\tau^\mathbf{E} + \mathbf{h^E}, \mathbf{z}_0^\mathbf{a}, \mathbf{z^I}]; \mathbf{c}_{\text{emb}}, \tau)$. When using these two specific conditions, the conditioning latents of the input video and albedo have their corresponding condition masks set to 1. For the remaining $70\%$ of the training steps, we use the default denoising function where we take input video tokens as conditions and predict the denoised relit video and albedo.

Our real-world auto-labeled data consists of RGB videos and corresponding albedos but lacks pairwise RGB videos and environment maps. The provided albedo $\mathbf{a}$ is used as a condition, and the RGB video is treated as the target relit video $\mathbf{I_E}$. Since the the original input video and the environment map information are unavailable for this dataset, their respective representations are set to $\mathbf{0}$: The environment map embedding $\mathbf{h^E}$ within the primary input array becomes $\mathbf{0}$, and the conditional input video representation $\mathbf{z^I}$ is dropped. The denoising function is thus $\mathbf{f}_\theta([\mathbf{z}_\tau^\mathbf{E} + \mathbf{0}, \mathbf{z}_0^\mathbf{a}, \mathbf{0}]; \mathbf{c}_{\text{emb}}, \tau)$. We adopt a $10\%$ probability of dropping conditions to enable classifier-free guidance [24].

## 5 Results

Before evaluating our method, we introduce baselines, metrics, and datasets. Then we conduct ablation studies on the principle of joint modeling and real-world auto-labeling of data. We refer to model implementation details in the Supplement.

**Baselines.** For the relighting task, we compare with the 2D methods DiLightNet [68], Neural-Gaffer [26], and DiffusionRenderer [39]. As the original DiffusionRenderer is built on Stable Video Diffusion [8], to isolate the effect of the base model and enable a fair algorithmic comparison, we re-implement DiffusionRenderer using the same Cosmos [49] backbone as our method, and adopt a data curation strategy closely following the original paper. We refer to this enhanced variant as DiffusionRenderer (Cosmos). For albedo estimation, we compare with IntrinsicImageDiffusion [32] and DiffusionRenderer [39].

**Metrics.** We apply the standard image metrics PSNR, SSIM, and LPIPS [74] to evaluate relighting and albedo estimation results. For relighting, we also conduct a user study to evaluate the perceptual quality of the produced results. Following prior works [32, 26, 39], we counteract scale ambiguity in both relighting and albedo estimation by a three-channel scaling factor using least-squares error minimization before computing the metrics.

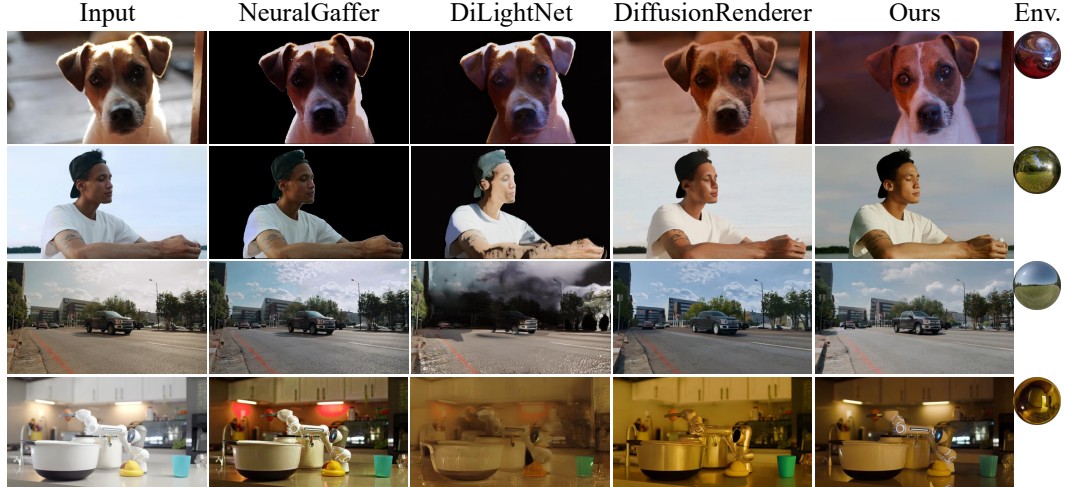

Figure 4: **Qualitative comparison on in-the-wild data.** Our method generates more plausible results than the baselines, with higher quality and more realistic appearance.

Table 1: Quantitative evaluation of relighting, including a user study, where "Ours preferred" indicates the preference over the baselines. A preference over the $> 50\%$ indicates Ours outperforming baselines.

| | SyntheticScenes | | | MIT multi-illumination | | | |
| | PSNR ↑ | SSIM ↑ | LPIPS ↓ | PSNR ↑ | SSIM ↑ | LPIPS ↓ | Ours preferred |
|---|---|---|---|---|---|---|---|
| DiLightNet [68] | 19.13 | 0.584 | 0.319 | 15.79 | 0.539 | 0.368 | 92% ± 8% |
| Neural Gaffer [26] | 19.17 | 0.638 | 0.263 | 17.87 | 0.683 | **0.241** | 84% ± 2% |
| DiffusionRenderer [39] | 24.17 | 0.768 | 0.217 | 16.99 | 0.582 | 0.342 | 96% ± 4% |
| DiffusionRenderer (Cosmos) | 26.61 | 0.841 | 0.222 | 17.29 | 0.622 | 0.355 | 88% ± 8% |
| Ours | **26.97** | **0.847** | **0.190** | **20.76** | **0.749** | 0.251 | n/a |

**Datasets.** We have curated a high-quality synthetic dataset named *SyntheticScenes* for quantitative and qualitative evaluation of both relighting and albedo estimation tasks. This dataset includes 3D assets from PolyHaven [67] and Objaverse [16], ensuring that no assets were used in the training of our method or any of the baseline methods. *SyntheticScenes* features 40 scenes, each constructed with a plane textured with random physically-based materials. Each scene is rendered into a 57-frame video sequence under four different lighting conditions, incorporating orbiting camera motions and rotating environment lighting. We cycle through the lighting conditions, selecting one video as the input, and the following as the relighting target, resulting in four different relighting tasks per scene. Additionally, we utilize the test set of *MIT multi-illumination* benchmark [48], which includes 30 scenes under 25 different illuminations. In this dataset, images are captured sequentially under adjacent flashlight illuminations. For our relighting tasks, the image captured under the $i$-th lighting condition serves as input, while the image captured under the $(i + 12)$-th lighting condition is utilized as the target illumination.

## 5.1 Evaluation

**Qualitative comparison.** In Figure 3 and Figure 4, we present a comparative analysis of our method against recent state-of-the-art relighting techniques: DiLightNet [68], NeuralGaffer [26], and DiffusionRenderer [39]. Our approach demonstrates superior performance, particularly in handling intricate shadows and inter-reflections. As illustrated in Figure 3, while DiffusionRenderer performs well on the synthetic dataset, it struggles to accurately represent complex materials—such as anisotropic surfaces, glass, and transparent objects—when utilizing G-buffers as an intermediate state, leading to suboptimal results. NeuralGaffer and DiLightNet do not produce accurate shadows and reflections, leading to poor relighting effects. Specifically, NeuralGaffer often makes very subtle, or no changes to the input video.

Figure 4 compares relighting results on in-the-wild data. For the first two object-centric scenes, we provided DiLightNet and NeuralGaffer with masks (we used DiLightNet's auto-masking, which is based on Segment Anything [29]) to produce more reasonable results. Notably, DiffusionRenderer struggles with fur, human skin, and car windows in the first three cases, as these are hard to represent

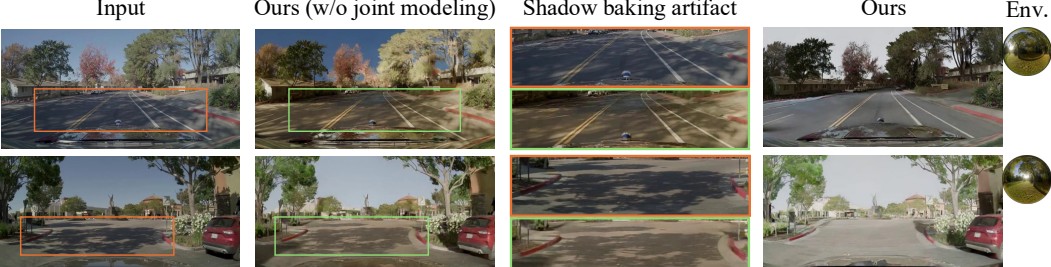

| Input | Ours (w/o joint modeling) | Shadow baking artifact | Ours | Env. |

Figure 5: **Ablation on joint modeling.** Relighting results on urban street scenes. The orange and green crops highlight regions where the pure relighting model (w/o joint modeling) clearly bakes shadows from the input image into the relit result. Our joint model correctly demodulates the shadows.

with simple G-buffers. It also misestimates the material of the plastic basin, leading to metal-like appearances. Overall, our relighting model is more general-purpose and can be used more effectively in diverse scenes, yielding comparatively more accurate and high-fidelity relighting results.

**Quantitative comparison.** In Table 1, we present a comparative analysis of our method against all baselines on both *SyntheticScenes* and the MIT multi-illumination benchmark [48]. DiffusionRenderer works well on *SyntheticScenes*, and the Cosmos version—a stronger base model—leads to improved fidelity. However, on the MIT multi-illumination dataset, DiffusionRenderer fails to model the more challenging materials using only the G-buffers, and even the Cosmos version cannot bridge this gap.

An outlier is observed in the LPIPS score for NeuralGaffer on the MIT multi-illumination dataset. As mentioned before, NeuralGaffer makes minimal changes to the input and performs poorly, but still achieves the lowest LPIPS score. We speculate that LPIPS considers the images sufficiently similar due to the relatively small deviation in light direction for this dataset.

**User study.** We conducted a user study to evaluate the perceptual quality of our relighting method compared to the baseline methods on the MIT multi-illumination dataset. Participants were shown a ground-truth relit image alongside two relighting results–one generated by our method and the other by a baseline model, with the order randomly shuffled. They were asked to select the result that more closely resembled the ground truth, considering aspects such as transparency, shadows, and reflections. For each sample pair, 11 users made a binary selection, and majority voting determined the preferred method for each comparison. We repeated the study three times, involving a total of 33 users, and report the average percentage of samples where our method is preferred over baselines in Table 1, along with the standard deviation for the three experiments. Our results are consistently preferred, strongly outperforming the other baselines.

**Albedo estimation.** While our primary focus is on relighting, we also evaluate the quality of our albedo estimates compared to the baselines in Table 2. While the Cosmos version of DiffusionRenderer has a slight edge, our method performs or on par with previous work in all metrics.

Table 2: Quantitative evaluation of albedo estimation on *SyntheticScenes*.

| | *SyntheticScenes* | | |
| --- | --- | --- | --- |
| | PSNR ↑ | SSIM ↑ | LPIPS ↓ |
| IntrinsicImageDiffusion [32] | 16.41 | 0.543 | 0.395 |
| DiffusionRenderer [39] | 26.22 | 0.837 | 0.166 |
| DiffusionRenderer (Cosmos) | **28.56** | **0.911** | **0.131** |
| Ours | 28.07 | 0.877 | 0.167 |

## 5.2 Ablation Study

**Ablation on joint modeling.** We ablate our joint modeling approach with quantitative and qualitative results in Table 3 and Figure 5. We compare our model to a pure relighting model (without joint modeling), whose denoising function is defined as $\mathbf{f}_\theta([\mathbf{z}_\tau^E + \mathbf{h}^E]; \mathbf{z}^I, \tau)$. Both models are trained exclusively on synthetic data. The quantitative results in Table 3 reveal that our joint model slightly improves relighting quality compared to the ablated model. The table also shows that providing ground truth albedo as input to our joint model further enhances quality, highlighting the model's capacity to leverage albedo information effectively. For a qualitative assessment, Figure 5 compares the results on urban street scenes outside the training distribution, where urban street scenes typically exhibit strong shadows under sunlight. The joint modeling approach demonstrates superior generalization, characterized by reduced false shadowing, as clearly shown in the figures, whereas the model without joint modeling largely bakes the shadow from the input.

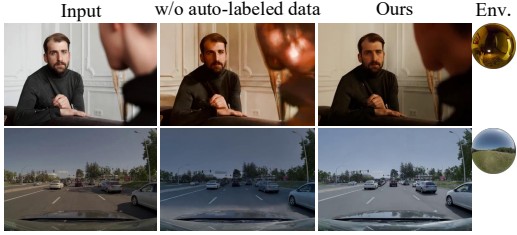

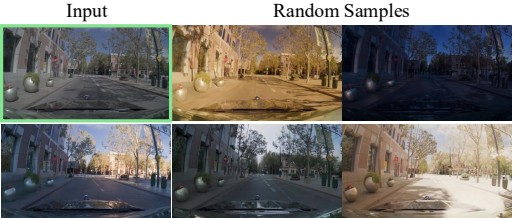

Figure 6: **Ablation on real-world auto-labeled data.** Although the dataset is sparsely labeled, it helps the model generalize to natural images.

Figure 7: **Application for data augmentation.** The top left image with green outline is the input image.

Table 3: Ablations on joint modeling designs evaluated on the *SyntheticScenes* dataset.

| Ablated Versions | PSNR ↑ | SSIM ↑ | LPIPS ↓ |
|---|---|---|---|
| Ours (w/o joint modeling) | 26.42 | 0.842 | 0.191 |
| Ours | 26.97 | 0.847 | 0.190 |
| Ours (w/ GT albedo) | **27.15** | **0.857** | **0.181** |

Table 4: User study of relighting on *StreetScenes*. Each row compares an ablated variant against the base version (Ours w/o auto-labeled data), reporting percentage of samples where users prefer the base version.

| Ablated Versions | Ours (w/o auto-labeled data) preferred |
|---|---|
| Ours (w/o auto-labeled data, w/o joint modeling) | 68% ± 14% |
| Ours (w/ auto-labeled data) | 45% ± 8% |
| Ours (w/o auto-labeled data, w/ inference-time albedo) | 54% ± 13% |

**Ablation on real-world automatically labeled data.** We ablate the usefulness of our automatically labeled real-world data in Figure 6. The model trained on solely the synthetic and MIT multi-illumination datasets (center column) exhibits noticeable artifacts for out-of-focus regions and outdoor scenes. This is expected, as both datasets have very limited depth of field and primarily feature indoor scenes or lack sky-like backgrounds. As shown in the figure, even though the real-world auto-labeled dataset contains only a subset ($\mathbf{I}$, $\mathbf{a}$) of the video labels, it significantly helps the model generalize to out-of-domain data, thereby improving image quality.

**User study.** We conducted a user study to evaluate the perceptual impact of our design choices on relighting quality, using 19 urban street scenes. For each scene, participants were shown a reference image and two relit videos produced by ablated versions of our method (randomized order). They were asked to select the video with more realistic lighting, focusing on aspects such as shadows and reflections. As before, each comparison was evaluated by 11 participants through binary selection, and the preferred result was determined by majority vote. We repeated the study three times with a total of 33 participants and report the average preference rate and the standard deviation in Table 4.

When trained only on multi-illumination data (w/o auto-labeled data), the joint modeling variant is strongly preferred over direct relighting. Adding real-world auto-labeled data further improves perceptual quality, consistent with our qualitative and quantitative findings. When comparing models with and without estimated albedo at inference time, the larger standard deviation in user preference suggests the perceptual quality is comparable. This indicates our method does not strictly rely on estimated albedo at test time.

## 5.3 Application: Illumination Augmentation

Our model's strong generalization capability enables effective data augmentation for scenarios such as driving and robotics scenes. We test our model under conditions without environment maps and with different random seeds. Figure 7 shows five random illuminations of one input scene. Our model generates diverse data, including nighttime and dusk scenes, demonstrating that it accurately models the illumination distribution and can sample realistic relighting results under varying lighting conditions.

## 6 Discussion

To overcome data scarcity and the limitations of decoupled two-stage methods in realistic relighting, we present a novel framework UNIRELIGHT, that *jointly* models scene intrinsics and illumination. This approach enhances generalization, reduces error accumulation, and more effectively captures complex visual effects by implicitly reasoning about scene properties rather than relying on explicit G-buffers. Furthermore, when trained on real-world auto-labeled data, our model achieves state-of-the-art, high-quality, and realistic relighting results.

**Limitations and future work.** While our method performs well on diverse data, our model cannot handle emitting objects, such as toggling lights within scenes. Our design, which is focused on environmental lighting, does not extend to emittance effects, which remains an area of future research. Moreover, incorporating text-based relighting could significantly enhance usability. Future work could investigate the conditioning of lighting with natural language by incorporating text cross-attention in our DiT model, a feature currently not implemented. While our method can generate reasonable shadows, integrating our generative approach with explicit 3D information—such as traced shadow buffers—represents a promising direction for improving physical accuracy. Besides, in UNIRELIGHT, we incorporate the albedo to improve the model's generalization, and predicting additional properties could further enhance the model's understanding of scene structure and materials. However, memory requirements scale with the number of properties, making this approach expensive for high-resolution videos; it is worth exploring this direction further.

## Acknowledgments and Disclosure of Funding

The authors thank Tianshi Cao and Huan Ling for their insightful discussions that contributed to this project. This work was conducted at and supported by NVIDIA.

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

# A    Broader Impact

We present UniRelight, a generative framework that jointly estimates albedo and synthesizes relit videos from a single input, enabling diverse lighting manipulation across both synthetic and real-world scenes. This capability can support a range of applications, including creative content generation, visual effects, virtual production, and potentially data augmentation for training more robust computer vision models in domains such as robotics and autonomous driving.

As with all generative video models, UniRelight may reflect biases present in its training data. Such biases could lead to relighting results that fail to generalize to underrepresented scene types or lighting conditions. Furthermore, tools for lighting manipulation carry the risk of misuse, such as altering or misrepresenting visual content in sensitive contexts like surveillance or media.

We discourage the use of UniRelight in applications where relighting may contribute to misinformation, misattribution, or privacy violations. In human-centric use cases, we recommend careful dataset curation to ensure fair representation across skin tones, races, and gender identities. Practitioners are encouraged to critically assess and de-bias training data to mitigate unintended harms where appropriate.

# B    Experimental Details

## B.1    Implementation Details

We fine-tune our models based on `Cosmos-Predict1-7B-Video2World` [49], a pre-trained DiT video diffusion model.

The encoded latents $\mathbf{z^I}$, $\mathbf{z^a}$, $\mathbf{z^E}$, $\mathcal{E}(\mathbf{E}_{\text{ldr}})$, $\mathcal{E}(\mathbf{E}_{\text{log}})$, and $\mathcal{E}(\mathbf{E}_{\text{dir}})$ are all in $\mathbb{R}^{l \times h \times w \times C}$, where $C = 16$. We use $C_{\text{emb}} = 3$ as the dimension of the type embedding. Thus, the concatenated tokens have a channel dimension of $16 + 1 + (16 + 1) \times 3 + 3 = 71$, where each latent has an associated binary condition mask (added as an extra channel), and the lighting features ($\mathbf{E}_{\text{ldr}}$, $\mathbf{E}_{\text{log}}$, $\mathbf{E}_{\text{dir}}$) each include a condition mask to indicate whether they are provided.

We adopt an image-video co-training strategy and train the model in two stages. Firstly, we train with only synthetic data by mixing the image data (sampling one frame from the video data) and the video data in a ratio of $1 : 1$ for 15,000 iterations. Then we train with all data with random sampling, including synthetic video data, synthetic image data, real-world auto-labeled data, and MIT multi-illumination data, in a ratio of $8 : 1 : 3 : 2$ for 12,000 iterations. We augment the real-world auto-labeled data with random flipping.

All training is done with a batch size of 64, using the AdamW optimizer with a learning rate of $2 \times 10^{-5}$, with mixed-precision (BF16) training at a resolution of $480 \times 848$ pixels. The AdamW optimizer was employed with a weight decay of $0.1$. The exponential decay rates for the moment estimates $\beta$ are set to $0.9$ for the first moment and $0.99$ for the second moment, with $\epsilon$ at $1 \times 10^{-10}$. The total training of two stages takes around 4 days on 32 A100 GPUs.

During inference, we use 35 denoising steps. We do not apply classifier-free guidance (CFG), as we empirically found that inference without CFG yields more accurate and visually consistent results.

**Baseline configurations.**    Since DiLightNet [68] requires a text prompt per example, we use `meta/llama-3.2-90b-vision-instruct`[0] to generate a short prompt for each example in the datasets based on the first image in each clip with the instruction *"What is in this image? Describe the materials. Be concise and produce an answer with a few sentences, no more than 50 words."*

As each of the baselines generates videos in different resolutions, for UNet-based baselines, we run inference on the model with the video first resized to $486 \times 864$ and then center-cropped to a

---

[0]https://www.llama.com/

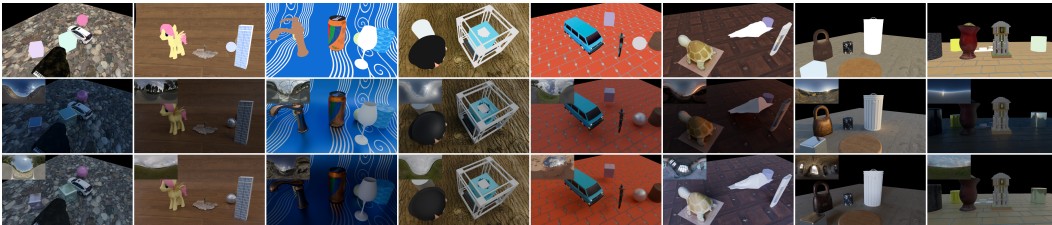

Figure 8: **Synthetic data visualization.** Randomly sampled example images are shown from our synthetic rendering data. The top-most images show albedo maps, while the bottom two rows display rendered scenes under two different illuminations with the corresponding environment maps.

Table 5: Quantitative comparison with IC-Light.

|  | PSNR ↑ | SSIM ↑ | LPIPS ↓ |
|---|---|---|---|
| IC-Light [73] | 18.08 | 0.834 | 0.096 |
| Ours | **23.19** | **0.901** | **0.079** |

Table 6: Evaluation of inference runtime cost.

|  | Runtime cost (seconds) ↓ |
|---|---|
| DiffusionRenderer [39] | 566.6 |
| DiffusionRenderer (Cosmos) | 780.0 |
| Ours | **445.5** |

resolution of $448 \times 832$; for our DiT-based model, we run inference on the model with the resized video with resolution of $486 \times 864$ and then center-cropped to $448 \times 832$ to align the results.

**Quantitative comparison configurations.** For quantitative evaluation, we apply background masks to the synthetic dataset to focus on foreground appearance. For the MIT multi-illumination dataset, we follow the dataset protocol and mask out light probes in all outputs before computing metrics.

## B.2 User Study Details

We conducted two user studies on Amazon Mechanical Turk to evaluate the perceptual quality of relighting results.

**MIT multi-illumination dataset** is a public benchmark with ground truth relighting. Participants were shown three images: a ground truth relit image and two relighting results—one generated by our method and one by a baseline model. Their task was to choose the result that more closely resembled the ground truth, considering attributes such as transparency, shadows, and reflections.

The exact instructions shown to participants were as follows:

> Carefully compare Image A, the Reference Image, and Image B. Your task is to determine which image (A or B) is more similar to the Reference Image.
>
> To make an informed decision, you may zoom in to examine the details. Pay close attention to aspects such as lighting, reflections, and shadows, as these can affect how natural the image appears.
>
> Once you have compared the images, select the one that best matches the Reference Image.
> ☐ Image A
> ☐ Image B

We evaluated 30 scenes from the test set, comparing our method against four baselines. The study was repeated three times with 11 unique participants in each run. In total, this resulted in $30 \times 4 \times 11 \times 3 = 3960$ individual comparisons.

**StreetScenes Dataset.** This dataset contains 19 urban street scenes without ground-truth relighting. Participants were shown a reference image along with two relit videos generated by different ablated versions of our method.

The instructions presented to participants were as follows:

> In this study, you will be shown a Reference Image and two videos – Video A and Video B – that changes the lighting of the scene. Your task is to watch both

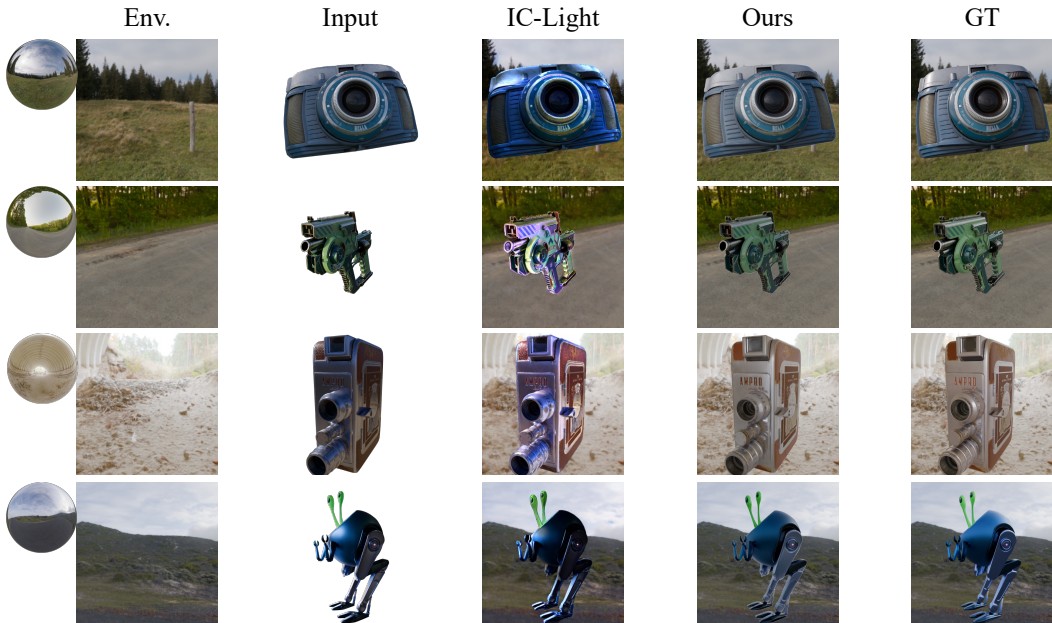

| Env. | Input | IC-Light | Ours | GT |
|---|---|---|---|---|

Figure 9: **Qualitative comparison with IC-Light [73].** We provide the environmental background used for IC-Light conditioning, with the reference environment ball on the left. Our method produces higher-quality and more accurate relighting results.

videos and choose which one (A or B) you think has more realistic shadows and reflections. To make an informed decision, you may zoom in to examine the details. Pay close attention to aspects such as lighting, reflections, and shadows, as these can affect how natural the image appears.

Once you have compared the videos, select the one that has more realistic lighting effects.

☐ Video A
☐ Video B

We evaluated 19 scenes, comparing a base version against two ablated versions of our method. The study was repeated three times with 11 unique participants in each run. In total, this resulted in $19 \times 2 \times 11 \times 3 = 1254$ individual comparisons.

### B.3 Synthetic data visualization

We show a synthetic data visualization in Figure 8. Each scene contains albedo videos, two environment maps, and pairwise videos rendered under the environment maps.

## C Additional Results

**Runtime cost.** We evaluate the inference runtime of our model on a 57-frame video at a resolution of $480 \times 848$. The overall inference time for performing 35 denoising steps, including VAE encoding and decoding, is 445.5 seconds, measured on a single A100 GPU.

To contextualize this cost, we compare our method against two baselines: DiffusionRenderer [39] and its Cosmos-based variant. All methods use 35 denoising steps for consistency. DiffusionRenderer is run at $448 \times 832$ resolution (slightly smaller than ours but divisible by 32 to fit its architecture) while the Cosmos variant is run at our native resolution.

For baseline methods, the total runtime is the sum of the inverse rendering and forward rendering durations. Notably, DiffusionRenderer requires five separate inverse rendering passes and one forward rendering pass per video, resulting in significantly higher computational cost. In contrast, our approach performs joint relighting and albedo estimation in a single pass and is correspondingly faster. Full timing results are shown in Table 6.

**Comparison with IC-Light [73].** We compare our method with the single-image relighting approach IC-Light [73] on object-centric synthetic data, as shown in Table 5 and Figure 9. Note that the

| Input | NeuralGaffer | DiLightNet | DiffusionRenderer (Cosmos) | Ours | GT | Env. |
|---|---|---|---|---|---|---|

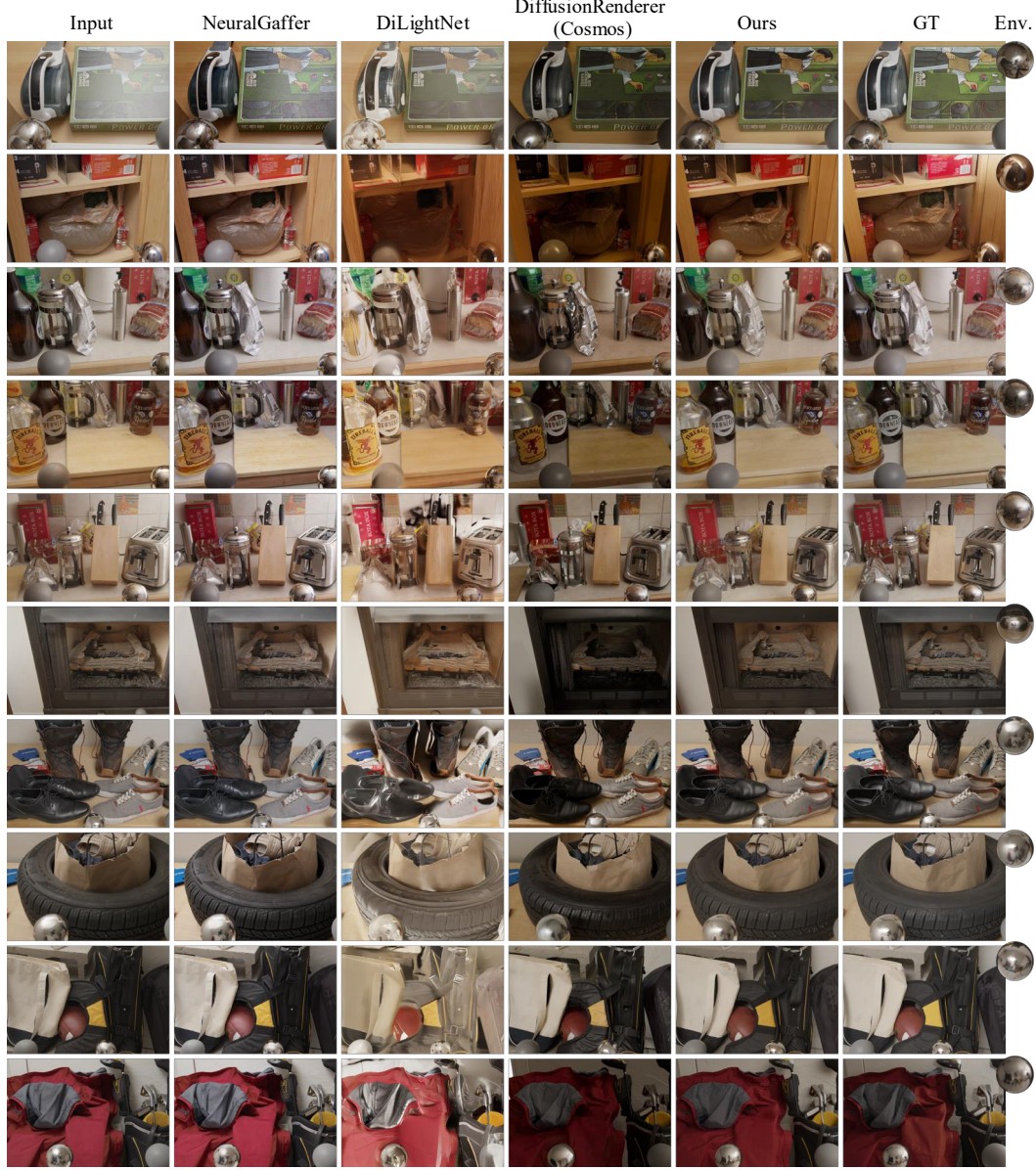

Figure 10: **Additional qualitative comparison on MIT multi-illumination dataset.** Our method consistently achieves more accurate relighting results than all baselines on the MIT multi-illumination dataset, demonstrating strong capability in relighting complex materials.

two methods follow different relighting formulations: IC-Light is designed for object relighting using background context as the primary cue—without access to an explicit environment map, while our method is conditioned on full HDR illumination, but is not specifically tuned for object-centric data.

Our method shows improved quantitative and qualitative performance. Since IC-Light relies on background appearance as its primary cue and has less information about the surrounding lighting, it may retain input-specific effects in its outputs—such as specular highlights and shadows, which can limit accuracy under novel lighting conditions. In contrast, our method produces more faithful relighting results, with sharper specular highlights, more realistic shadows, and improved visual fidelity.

**Additional qualitative comparison on the MIT multi-illumination dataset.** We provide additional qualitative comparisons on the MIT Multi-Illumination dataset in Figure 10. To ensure a fair comparison, we include results from our re-implemented version of DiffusionRenderer using the Cosmos backbone, which achieves higher visual fidelity than the original implementation. Our method

| Input | Relighting I | Relighting II | Relighting III |
|---|---|---|---|

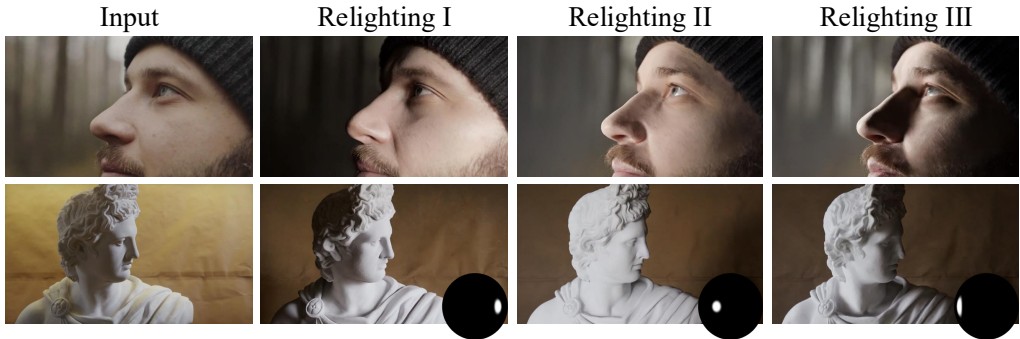

Figure 11: **Additional qualitative results under point-light illumination.** The bottom right of each column indicates the target lighting conditions. Our results show strong robustness of our method under point-light illumination.

| Input | Estimated Albedo | Relighting | Env. |
|---|---|---|---|

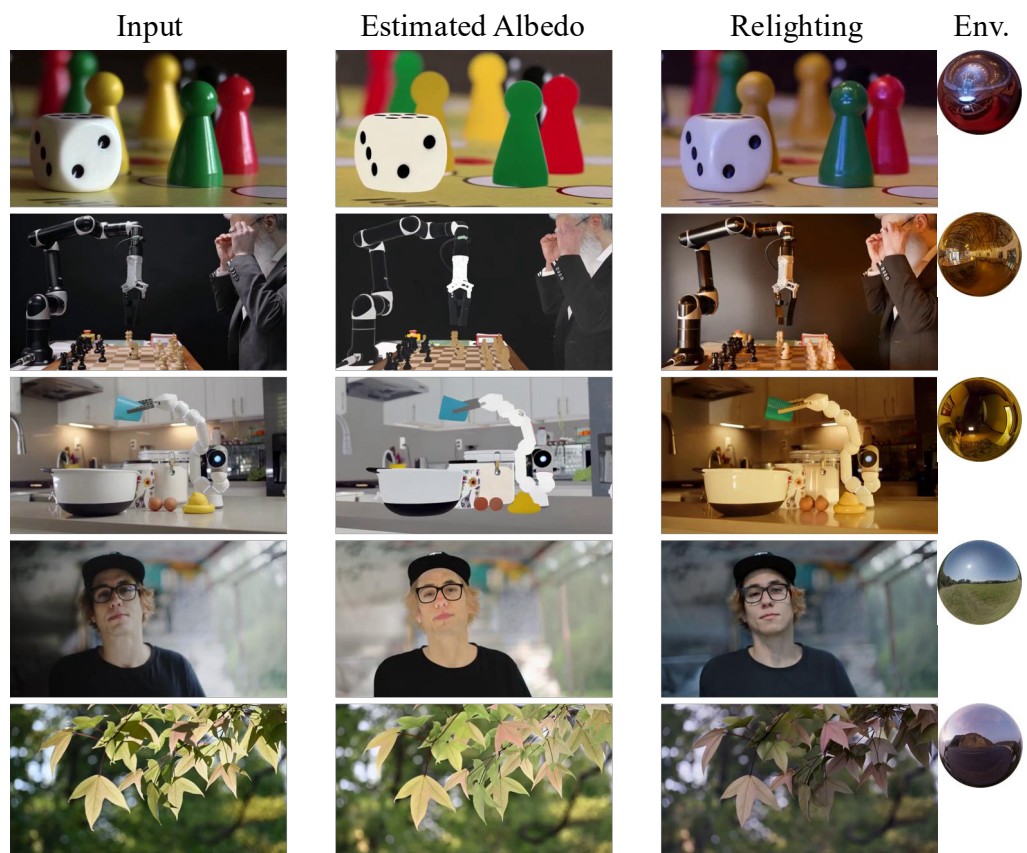

Figure 12: **Additional qualitative results on real scenes.** Our method provides high-quality albedo estimation and realistic relighting results.

consistently produces more accurate transparency, specular highlights, and shadows across scenes, demonstrating strong capability in handling complex materials and outperforming all baselines in visual quality.

**Additional qualitative results under point-light illumination.** We further evaluate the robustness of our method in extreme cases, such as point-light illuminations, which do not exist in our training data. As shown in Figure 11, our method produces high-quality relighting results, demonstrating the strong robustness and generalization capability of our method.

**Additional qualitative results on real scenes.** We present additional results on real scenes in Figure 12. Our method produces high-quality albedo and relighting results with realistic specular highlights and shadows under target lighting conditions.

