# OpenReview forum: "UniRelight: Learning Joint Decomposition and Synthesis for Video Relighting"
_NeurIPS.cc/2025/Conference — NeurIPS 2025 spotlight_

### Official Review · Reviewer_PiUf · 2025-06-22

**Clarity:** 3
**Significance:** 3
**Originality:** 2
**Rating:** 5
**Confidence:** 3

**Summary:**

The paper performs relighting by leveraging the implicit lighting knowledge captured in a video diffusion model. The key insight in this paper is borrowed from similar ideas in the past which performed joint estimation of different modalities (i.e. curvature, depth, semantics) to constrain the optimization of the diffusion model weights. The method trains using the known diffusion model cost functions with the extra albedo regularization term. Results exhibit convincing results on both qualitative (user survey) and quantitive.

**Questions:**

See weaknesses

**Ethical Concerns:**

["NO or VERY MINOR ethics concerns only"]

**Final Justification:**

The paper shows how video diffusion priors could be leveraged for re-lighting. The community will benefit from this contribution.

**Limitations:**

Authors have provided a limitations section.

**Paper Formatting Concerns:**

Nothing major.

**Quality:**

3

**Strengths And Weaknesses:**

Strength:
- Method is intuitive, simple and effective. Borrowing from joint-estimation approaches that are well studied and shown to work.
- Paper well organized with good set of experiments.

Weaknesses:
- One of the main concerns is while results are qualitatively good (seen in the supplemented video), no comparisons were made with classical inverse rendering approaches. Hallucinations of shadows which might appear convincing to the naked eye will theoretically score lower against classical accurate methods.
- Not a weakness per-se, and I'm also not asking for an experiment. But I am curious how much effect the video diffusion model prior had on obtaining these results vs. a image based diffusion model.

---

> ### Author Rebuttal · Authors · 2025-07-31
>
> We thank the reviewer for their valuable feedback, and appreciate that they recognize our contributions. We respond to the comments and questions below.
>
> **Comparison with classical approaches.** We agree that classical physically based rendering (PBR) methods can achieve high accuracy when high-quality 3D geometry and materials are available. However, in the absence of carefully authored scene content, methods like screen-space ray tracing (SSRT) often struggle to produce artifact-free shadows and reflections, as also noted in DiffusionRenderer [39].
>
> In this work, we focus on enabling high-quality relighting directly from video without requiring full scene reconstruction. We agree that integrating our generative approach with explicit 3D information—such as traced shadow buffers—is a promising direction for improving physical accuracy, and we will discuss this in the revision as future work.
>
> **Video diffusion model vs. Image diffusion model.** Video diffusion models have better capacities to model dynamic scenes and ensure temporally consistent results. This is especially important for effects like reflections and shadows, which may otherwise flicker, pop in/out or change shape abruptly when occluders enter or leave the frame. Please refer to our qualitative comparisons in the supplementary video, where we also include results from image-based baselines for comparison.

---

> > ### Comment · Reviewer_PiUf · 2025-08-01
> >
> > I thank the authors for their response.

---

### Official Review · Reviewer_a48D · 2025-07-02

**Clarity:** 4
**Significance:** 3
**Originality:** 3
**Rating:** 5
**Confidence:** 5

**Summary:**

The paper introduces a generative relighting method that takes an input image or video alongside with a target environment lighting and produces relit images or videos. The key idea is that instead of doing intrinsic image decomposition first and then shynthesis, as usually done in the literature, the authors propose to model the the decomposition and relighting task jointly by predicted albedo and relighting at the same time. The model is fine-tuned on a set of datasets, such as synthetic relightings, following DiffusionRenderer, real multi-illumination dataset, and auto-labeled real-world dataset.

Comprehensive experiments validate the idea and show impressive, extremely realistic results.

**Questions:**

## Method
* The key insight of the paper is that jointly training for albedo estimation improves the relighting quality. Would it further improve if also training for further material parameters, such as roughness, metallic, or for lighting maps, as shading map, maybe even for geometric maps, as normal/depth? Also, is this improvement bi-directional, i.e. the albedo estimation also gets improved?
* As Table 3 also shows, the model can utilize albedo information as conditioning. Would it be possible to use this conditioning for material editing applications?

## Related works
* Please double-check the references and cite the conferences instead of arxiv, where possible (e.g. 32 was published at CVPR2024).

## Writing
* Typo: L.284: performs *better* or ...

**Ethical Concerns:**

["NO or VERY MINOR ethics concerns only"]

**Final Justification:**

The paper introduces a novel idea of jointly modelling intrinsic image decomposition and relighting, which can provide valuable insight to the community.

**Limitations:**

yes

**Paper Formatting Concerns:**

No concerns.

**Quality:**

3

**Strengths And Weaknesses:**

## Strengths
* **S1** Very impressive results. The proposed method seems to achieve emerging understanding of more complex reflectance and refraction properties beyond the albedo (as also seen in Figure 3 and the bottle in row 3).
* **S2** Well-written, easy to follow.
* **S3** Clear and logical morivation, provides valuable insight for the community.


## Weaknesses
* **W1 - Key claim not ablated** One of the key claims of the paper is that joint modelling is beneficial for relighting; better than two-stage inverse+forward rendering. However, there is no ablation in the paper, which underlines this claim. The authors compare against DiffusionRenderer, which indeed follows the usual inverse and forward rendering pipeline; however, DiffusionRenderer was trained on a different set of datasets (did not use the MIT multi-illumination dataset and smaller auto-labeled real dataset). Therefore, it is not clear, whether the improvement indeed comes from the joint modelling, or rather from the better training data. The paper provides comparisons against the single-stage, direct relighting, but not against a variant, where first and image-conditional model estimates the albedo, then another image+albedo+lighting conditional model generates the relit images.

---

> ### Author Rebuttal · Authors · 2025-07-31
>
> We thank the reviewer for their valuable feedback, and appreciate that they recognize our contributions. We respond to the comments and questions below.
>
> **Joint modeling ablation.** Our joint modeling formulation encourages the model to learn internal representations of scene structure, which improves generalization to unseen domains and mitigates shadow baking artifacts. We include a quantitative ablation in Table 3, and qualitatively demonstrate in Fig. 5 and Supplementary Video (0:44–0:59) that joint modeling effectively decomposes shadows and generalizes to challenging in-the-wild scenes.
>
> **Comparison with DiffusionRenderer.** We re-implemented DiffusionRenderer using the same Cosmos backbone and trained it on the same synthetic and auto-labeled real-world datasets, denoted as DiffusionRenderer (Cosmos). DiffusionRenderer decomposes relighting into two stages: an inverse rendering step that estimates scene G-buffers, followed by a forward rendering step that synthesizes relit images conditioned on these estimates. This design introduces fundamental limitations: it is sensitive to errors in the predicted G-buffers (e.g., roughness) and struggles to capture complex material properties not encoded by G-buffers, such as transparency. We present quantitative comparisons in Tab. 1 and qualitative results in Supp. Fig. 2, showing that our joint modeling yields higher quality and greater flexibility when learning from diverse supervision sources.
>
> **Comparison with two-stages model.** We experimented with an “image+albedo+lighting-conditional relighting model”. In practice, low-capacity versions of this design often either bias toward “flat” outputs resembling the input albedo, or suffer from “shadow baking” where shadows in the input image are preserved in the output. These limitations motivated our design: we increase model capacity via token-wise concatenation and explicitly model the joint distribution of scene intrinsics and relit appearance. Our single-stage formulation also enables faster inference compared to two-stage pipelines.
>
> **Predicting more material properties.** We agree that this is a promising direction, and predicting additional properties could further improve the model’s understanding of scene structure and materials. Currently, memory requirements scale with the number of properties, making this expensive for high-resolution videos. We will further discuss it in the revision and leave it as a direction for future work.

---

> > ### Comment · Reviewer_a48D · 2025-08-04
> > **All concerns resolved**
> >
> > I thank the authors for the provided details, which resolved all my concerns; thus, I will raise my score to "Accept".

---

### Official Review · Reviewer_tfJ3 · 2025-07-02

**Clarity:** 4
**Significance:** 3
**Originality:** 3
**Rating:** 5
**Confidence:** 4

**Summary:**

The paper introduces a video relighting model that can be conditioned on a target environment map. Prior work in video modeling showed that video models can learn to produce better motion if they are trained to output an auxiliary output connected to the motion (like optical flow), and this paper builds on top of that idea and outputs an albedo estimate of the video, in hopes that will improve the model's performance in video relighting. The authors use a mixture of synthetic data, and real-world data with and without GT.

**Questions:**

If predicting albedo improves performance, then why stop there? Wouldn't outputting material property also be beneficial? It seems that the ideal architecture would be outputting all the G-buffers like DiffusionRenderer does, and also output an RGB relighting video along it. Since the authors already reproduced DiffusionRenderer, this can be a much stronger method. I don't expect the authors to do this for the rebuttal, but it could potentially make the paper much stronger.

Line 145, encoding the ray maps with the video encoder seems unnatural. Why not only downsample it bilinearly? do we know if those ray maps are within distribution for the video encoder?

It appears in the results that the method is strongest relative to the baselines in scenes with objects that are semi-transparent or reflective, would it be possible to construct a dataset that makes the method completely stand out relative to the baselines and show off its strengths?

A quantitative dataset ablation, that shows the benefits of having each dataset can be interesting. Like for the synthetic dataset, MIT multi-illumination dataset, and the auto-labeling. And how the performance is affected on the test split of each dataset. I ask for this because it is not clear to me how much each dataset is helping the performance on the other datasets.

**Ethical Concerns:**

["NO or VERY MINOR ethics concerns only"]

**Final Justification:**

The paper introduces an interesting idea of outputting intrinsics as a pseudo-task to improve relighting, and shows positive results, so I recommend accepting the paper

**Limitations:**

yes

**Paper Formatting Concerns:**

No formatting concerns

**Quality:**

3

**Strengths And Weaknesses:**

Strengths:
- The paper's idea is sound. The proposed hypothesis is: would outputting auxiliary information related to lighting improve the performance? and the paper goes on to validate the hypothesis and show that it is true

- The relighting performance is consistent for single images with rotating environment maps, and also addresses shadows and reflections nicely

- The work manages to incorporate real world datasets into the training, which is nice since most prior relighting work relies exclusively on synthetic data

Weaknesses:
- I'm surprised that in Table 3, the gain of modeling the albedo is actually marginal. Isn't that the entire thesis of the paper? Also, I'd like to see similar ablation (w/o the GT albedo row) on the other used datasets like MIT multi illumination dataset. Restricting the results to the SyntheticScenes is odd

- DiffusionRenderer (Cosmos) has a better performance in outputting albedo (indicating that outputting the other buffers are beneficial), so this is begging the question, why not build on top of that and output the relit RGB as another "buffer"


Side comments:
- NeuralGaffer is trained on objects only, so it makes sense that it would behave oddly on scenes
- There are some multiview relighting papers that could be relevant to the related work (since prior work that relights multiple frames at the same time are limited). For example, SimVS (Trevithick et. al 2025) and generative multiview relighting (Alzayer et. al 2025)
- line 99: following the EDM —> following EDM


Trevithick, Alex, et al. "Simvs: Simulating world inconsistencies for robust view synthesis." Proceedings of the Computer Vision and Pattern Recognition Conference. 2025.

Alzayer, Hadi, et al. "Generative multiview relighting for 3d reconstruction under extreme illumination variation." Proceedings of the Computer Vision and Pattern Recognition Conference. 2025.

---

> ### Author Rebuttal · Authors · 2025-07-31
>
> We thank the reviewer for their valuable feedback, and appreciate that they recognize our contributions. We respond to the comments and questions below.
>
> **Joint modeling ablation.** Our joint modeling formulation encourages the model to learn internal representations of scene structure, which improves generalization to unseen domains and mitigates shadow baking artifacts. When evaluated on the in-domain synthetic test set (Table 3) or the MIT benchmark (PSNR 18.33 vs. 18.09), the gain from joint modeling may appear modest. However, its benefits are more evident in real-world out-of-domain scenarios. As shown in Fig. 5 and Supplementary Video (0:44–0:59), our joint modeling version effectively decomposes shadows and generalizes to challenging in-the-wild scenes. We will clarify this in the paper and add additional qualitative examples in the supplement.
>
> **Treating relit RGB as another “buffer”.** While relighting may appear structurally similar to intrinsic prediction, we empirically observed that simply adding relit RGB as another output buffer often leads to “shadow baking”, where shadows in the input are directly copied into the output. This motivates our design choice: we increase model capacity via token-wise concatenation and explicitly model the joint distribution of scene intrinsics and relit appearance. This encourages the model to disentangle shading effects and produce controllable, high-quality relighting.
>
> **Predicting more material properties.** We agree that this is a promising direction, and predicting additional properties could further improve the model’s understanding of scene structure and materials. Currently, memory requirements scale with the number of properties, making this expensive for high-resolution videos. We will further discuss it in the revision and leave it as a direction for future work.
>
> **Ray maps encoding.** Compared to downsampling, encoding the ray maps can preserve more details and align with other conditions in latent space during the denoising process.
>
> **Dataset usage.** We agree that constructing a more specific dataset, including semi-transparent or reflective objects, can help our model be more robust in these scenarios. Meanwhile, our auto-labeled data also helps our model generalize to more data domains, such as skin and fur (Fig. 4), which are all very challenging domains.
>
> **Dataset ablation.** We report an ablation on the auto-labeled dataset in Fig. 6 and Tab. 4. While we agree that a full dataset-level ablation would provide additional insight, it is currently prohibitive due to the high computational cost of training large-scale video diffusion models.

---

> > ### Comment · Reviewer_tfJ3 · 2025-08-06
> >
> > Thanks for the reviewers for answering the questions. I will raise my recommendation to "accept," although I would like to note it would make the paper stronger to answer the questions in the rebuttal with experiments in the final paper, as opposed to high level discussions.

---

### Official Review · Reviewer_QAPg · 2025-07-03

**Clarity:** 3
**Significance:** 4
**Originality:** 3
**Rating:** 5
**Confidence:** 4

**Summary:**

The paper proposes UniRelight, a video-diffusion framework that jointly estimates an albedo map and synthesises a relit video. By concatenating latent codes for albedo and lighting and denoising them together, the model avoids the error accumulation typical of two-stage pipelines, while allowing faster processing.
To overcome the scarcity of multi-illumination data, the authors want to release a large synthetic dataset for training and a validation dataset SyntheticScenes, and a auto-label 150 k Internet video clips with albedo maps generated by their own inverse-rendering approach. Experiments show higher PSNR / LPIPS and a strong user-study preference over baselines such as DiffusionRenderer and Neural Gaffer and DiLightNet. Ablations confirm the benefits of the joint formulation and the auto-labelled real-video pre-training.

**Questions:**

-  Probe robustness – How much do PSNR / LPIPS drop if the HDR probe is rotated, clamped, or replaced by a point-light?
-  Auto-label quality – What is the albedo noise level, and how does training without labels change results? Did you filter the datasets somehow for quality? Please add a few dataset visuals.
-  Geometry handling – Have you tested on shots with strong parallax or moving occluders?
-  Efficiency – What FPS and GPU memory does UniRelight need at 720p, 1080p, and 4 K, and how does that compare to DiffusionRenderer?
-  Synthetic-set coverage – Which lighting types (point, HDR dome, emissive) are in SyntheticScenes, and how were they sampled? Include representative examples.

**Ethical Concerns:**

["NO or VERY MINOR ethics concerns only"]

**Final Justification:**

I thank the authors for their response, which addresses my questions. I will leave my rating as set to accept.

**Limitations:**

Limitations are discussed in the manuscript.

**Paper Formatting Concerns:**

None.

**Quality:**

3

**Strengths And Weaknesses:**

Strengths

•	Single-pass joint decomposition + relighting elegantly unifies intrinsic estimation and light transport, giving visibly cleaner results than two-stage pipelines and simplifying the model architecture, which also leads to faster inference.

•	Temporal consistency is highlighted through a video-diffusion backbone, avoiding flicker (l. 109). However, the benefit over previous work could be shown in more detail with direct comparisons to DiffusionRenderer or DiLightNet.

•	The hybrid training strategy (l. 54) improves generalisation, though is only roughly explained in (l. 198-214). Further details are only added in the supplementary material and should be perhaps added in the main document.

•	Thorough evaluation shows strong baselines, standard metrics, user studies, and ablations.

•	A clear discussion of limitations (l. 332) aids the exploration of future work.


Weaknesses

•	Dependence on an HDR environment probe at inference; robustness to probe errors or indoor spotlights is untested.

•	No explicit geometry reasoning means large view changes or strong occlusions may degrade results.

•	Quality of auto-labels is not quantified; the effect of noisy albedo on final quality is unclear. It is also not fully clear how the overall auto-labelled dataset looks on a larger scale, as almost no qualitative figures are provided.

•	The run-time footprint (FPS, memory) is only briefly mentioned; practical deployment is uncertain and still far from real-time capabilities needed for adaptive resimulation frameworks or similar approaches.

•	Simulation datasets are neither qualitatively nor quantitatively shown. They may lack complex lighting, such as moving point sources or emissive objects, and the lighting paths—including how they are sampled—are not described.

---

> ### Author Rebuttal · Authors · 2025-07-31
>
> We thank the reviewer for their valuable feedback, and appreciate that they recognize our contributions. We respond to the comments and questions below.
>
> **Probe robustness.** Conditioning our model on light-probes assumes distant lighting, which is indeed a limitation. We note that our model often produces plausible results for indoor scenes, but do not expect it to scale to extreme cases such as local spotlights. Our training data includes HDR environment maps with randomized rotations and intensity, enabling the model to follow the control of environment map condition, and generalize well diverse probes. In practice, we observe stable performance under rotated or clamped probes. We can add results with environment maps with a single lit pixel/OLAT config in the revision.
>
> **Auto-label quality visualization.** We re-implemented DiffusionRenderer [39] using the Cosmos backbone [46] to label the G-buffers, and achieved high-quality estimation on diverse scenes. We appreciate the suggestion and will include additional visualizations of the auto-labeled dataset in the revised supplementary material. While the labels are “pseudo-groundtruth”, Fig. 6 shows that training on this data improves relighting quality, demonstrating the value of large-scale real-world data.
>
> **Synthetic dataset visualization.** Our synthetic dataset was rendered using around 2,000 HDR environment maps. We use a path-tracer to render video clips under randomized lighting rotations, intensities, and color temperatures. We will show some representative examples and include more details of our synthetic data generation in the revision.
>
> **Geometry handling.** Our model implicitly handles occlusions and synthesizes cast shadows as demonstrated in several scenes (e.g., tree shadows) shown in Supplementary video 0:44~0:59, Fig. 5, and Supp. Fig. 3. However, performance may degrade with thin structures or fast-moving occluders. Incorporating 3D geometry-aware priors (e.g., traced shadows from estimated depth or meshes) is a promising direction for future work.
>
> **Efficiency.** During inference, UniRelight uses more memory (37.9 GB @ 480×848 pixels x 25 frames) than Diffusionrenderer (32.0 GB @ 448×832 pixels x 25 frames). This is due to the joint modeling of albedo and relighting in a single denoising pass. Our inference is faster than DiffusionRenderer, as shown in Supp. Tab. 2, thanks to the single-pass architecture. We acknowledge that runtime and memory remain deployment challenges. As future work, we plan to adopt memory-efficient architectures such as Framepack [1] and explore model distillation. While 1080p and 4K are currently infeasible memory-wise using the Cosmos backbone on current GPUs, we leave acceleration and support for higher resolutions as future work.
>
> [1] Zhang, Lvmin and Agrawala, Maneesh. “Packing Input Frame Contexts in Next-Frame Prediction Models for Video Generation” Arxiv 2025.

---

> > ### Comment · Reviewer_QAPg · 2025-08-05
> > **All Concerns addressed**
> >
> > I thank the authors for their response, which addresses my questions. I will leave my rating as set to accept.

---

### Decision · Program_Chairs · 2025-09-17

**Decision:**

Accept (spotlight)

**Comment:**

The reviewers all appreciate the contributions of this work and recommend acceptance. The AC and SAC also found the joint denoising of scene albedos and relit outputs to be a significant contribution that overcomes the limitations of previous approaches and produces compelling results. A decision was therefore made to accept this paper for spotlight presentation.